# Rare and highly destructive wildfires drive human migration in the U.S.

Kathryn McConnell [1,2] ✉, Elizabeth Fussell [1,3], Jack DeWaard [4,5], Stephan Whitaker [6], Katherine J. Curtis [7], Lise St. Denis [8], Jennifer Balch [8] & Kobie Price[9]

The scale of wildfire impacts to the built environment is growing and will likely continue under rising average global temperatures. We investigate whether and at what destruction threshold wildfires have influenced human mobility patterns by examining the migration effects of the most destructive wildfires in the contiguous U.S. between 1999 and 2020. We find that only the most extreme wildfires (258+ structures destroyed) influenced migration patterns. In contrast, the majority of wildfires examined were less destructive and did not cause significant changes to out- or in-migration. These findings suggest that, for the past two decades, the influence of wildfire on population mobility was rare and operated primarily through destruction of the built environment.

In recent decades, wildfire destruction of the built environment has grown dramatically, posing a growing threat to human settlements across the U.S.[1,2]. This trend is driven in part by changes in wildfire patterns, with records showing increases in total acres burned, number of large fires, and length of fire weather season[3–6]. Models project that, under climate change, the potential for very large fires will increase in the coming decades[7]. Concurrent to the rise in wildfire frequency and severity, the number of people living in high fire risk regions has increased, with substantial population and housing growth in areas in close proximity to or intermixed with wildlands[2,8]. Consequently, an increasing number of dwellings and their residents are exposed to wildfires.

The growing scale of wildfire destruction to buildings has the potential to impact human mobility patterns, yet little is known about the relationship between wildfire destruction and human migration. Wildfire-related mobility is notably absent in systematic reviews of environmental migration literature[9–12], a gap highlighted by the Intergovernmental Panel on Climate Change[13].

Previous studies of other environmental hazards indicate that climate-migration relationships vary widely in their direction and magnitude, differing between hazard types, as well as by the geographic, social, and economic contexts of affected populations[9,11–13].

While in some studies, weather and climate extremes are associated with heightened out-migration, in others, hazards cause minimal impact and relative immobility[14–16]. Given this heterogeneity of hazard-mobility relationships, researchers working in this field do not expect consistent or simple "push" effects, in which residents necessarily move away from hazardous places[9]. Instead, environmental migration scholarship investigates a range of different migration and non-migration responses to environmental change, with special attention to distinct hazards and the thresholds at which migratory effects occur[11,12].

Within existing environmental migration research, the most relevant studies for comparison to wildfire are those that examine sudden-onset hazards, such as floods, tsunamis, and hurricanes. Compared to the stronger migratory effects of slow-onset environmental changes such as drought or precipitation anomalies, sudden-onset events are more often found to have null or even negative effects on migration. Prior research has suggested that this relative immobility results from financial liquidity constraints, as household wealth is destroyed by the event, constraining funds needed to move[10–12,17]. Some describe those experiencing this form of involuntary immobility as "trapped populations"[18]. While certain households and populations may have more limited capability to migrate, others may be able to

[1]Population Studies and Training Center, Brown University, Providence, RI, USA. [2]Department of Sociology, The University of British Columbia, Vancouver, BC, Canada. [3]Institute at Brown for Environment and Society, Brown University, Providence, RI, USA. [4]Population Council, New York, NY, USA. [5]Center for Studies in Demography and Ecology, University of Washington, Seattle, WA, USA. [6]Federal Reserve Bank of Cleveland, Cleveland, OH, USA. [7]University of Wisconsin—Madison, Madison, WI, USA. [8]Earth Lab, University of Colorado Boulder, Boulder, CO, USA. [9]University of Minnesota—Twin Cities, Minneapolis, MN, USA. ✉e-mail: kathryn.mcconnell@ubc.ca

move but desire to remain in place. Such voluntary immobility in the face of intensifying environmental hazards can be due to a range of factors, such as the strength of place-embedded social and economic networks, the draw of local environmental amenities, and residents' ability to mitigate localized hazard exposure[19–21].

In the context of wildfire, immobility dynamics may play out in a number of ways. Recent research has linked rising housing costs in urban cores of California to the expansion of less costly housing development in exurban and rural wildfire-prone places[22]. This trend suggests that some residents of fire-prone places may have limited ability to move away from hazards due to regional housing affordability constraints. Other researchers have emphasized the pull of environmental amenities, drawing residents to voluntarily live in fire-prone places[8,20]. This research indicates that immobility may be a prevalent response to wildfire, and one that should be given equal attention as environmentally-linked mobility[14].

While immobility is often documented in response to sudden-onset hazards, select studies on very extreme events—such as Hurricane Katrina in the U.S. Gulf Coast, Hurricane Maria in Puerto Rico, and the Indian Ocean Tsunami in Indonesia—have also illustrated clear patterns of heightened post-disaster out-migration, or, displacement[23–26]. Collectively, these findings illustrate that sudden-onset environmental shocks may cause a continuum of migratory effects, ranging from immobility to large-scale out-migration. Such variability speaks to the importance of investigating the impacts of hazards across a spectrum of severity levels, ranging from the most extreme events to those less severe but more common hazards.

Scholars have recently begun studying wildfire-related mobility, for instance through investigation of migration intentions related to wildfire and wildfire smoke[27,28] and household decisions to remain in place after wildfire[29]. Several quantitative studies have documented patterns of temporary evacuation after major wildfires and long-term migration following subsets of disaster-level fires. These studies report heterogeneous effects across different events, in some cases documenting minimal changes to migration patterns, while in others showing heightened out-migration and reduced in-migration[20,30–32].

To provide greater insight and more generalizable knowledge of wildfire-mobility dynamics, we investigate patterns of out- and in-migration following highly destructive wildfires that occurred in the contiguous U.S. over more than two decades. Building on Hoffman et al.'s distinction between direct and indirect environmental migration drivers[12], our study tests two hypotheses on the relationship between wildfires and human mobility, positing that wildfires influence migration patterns through two broad pathways: (1) through direct damage to the built environment, and (2) through indirect mechanisms other than impacts to the built environment.

In the first proposed pathway, wildfires drive migration through their effects on the built environment, whereby destroyed structures result in out-migration via housing and other infrastructure loss. We define "structures" broadly to include residential, commercial, outbuilding, and mixed-use buildings[33]. We interpret heightened out-migration following highly destructive wildfires as evidence of damage-driven migration effects, akin to hazard-driven displacement. Our data show that the number of structures destroyed per wildfire has a long right skew, in which a small number of fires caused an outsized proportion of damage[33]. Given this distribution, we anticipate that wildfire effects on migration via the built environment would likely be non-linear, wherein as the number of structures destroyed grows, the number of out-migrants will increase at an increasing rate as local areas are unable to accommodate residents whose residences were destroyed. Such non-linear effects have been documented in the cases of extreme temperature variations[34] and rainfall[17], among others. Thus, our first hypothesis is that damage-driven out-migration will be greatest in areas experiencing high levels of fire-related destruction in the event period.

In the second proposed pathway of wildfire-driven mobility, we hypothesize that wildfire may influence migration indirectly through a range of other mechanisms that are distinct from direct displacement via structure loss. These indirect mechanisms can be broadly characterized as changes in residential preferences of where to live and/or residents' capabilities to realize these preferences[12,21,35]. For instance, residential preferences may be influenced by wildfire-related changes to natural amenities, air quality, local economic conditions, and perceptions of future fire risk and potential losses. Residents' mobility capabilities may also change, for instance through impacts to household finances or reduced access to homeowner's insurance[19]. While we are unable to parse individuals' migration motives with our data, we broadly test for the presence of indirect wildfire effects by examining migratory responses to wildfires in places experiencing lesser impacts to the built environment and at later time periods relative to the event.

If indirect mechanisms are driving wildfire-related migration, we would expect to observe the following changes. First, out-migration will increase in areas that experience lower levels of wildfire destruction, in particular following events in which too few structures are destroyed to directly displace a large number of residents. Second, out-migration will be elevated in burned areas during the temporal period beyond the disaster event, for example several years afterward. In this period, structure loss is unlikely to be the motivation, but other changes caused by the fire may influence residential preferences and/or capabilities, resulting in migration. Third, in-migration to fire-affected places will decline as potential in-migrants seek to avoid the fire-affected destination. Any of these changes would support the hypothesis that indirect mechanisms based on changes in residential preferences and/or mobility capabilities drive wildfire-related migration.

The null hypothesis to the direct and indirect hypotheses of wildfire-induced mobility is, conversely, immobility: migration flows into and out of fire-affected areas will not change in response to wildfire events. Immobility is informed by residents' desires to remain in place or to move, their capabilities to realize those aspirations[14,16,19], and protections, such as home hardening or firefighting resources, that allow people to remain in place. As such, immobility in the face of wildfire destruction could reflect voluntary immobility of residents' desire to remain living in fire-prone places, but also may reflect certain populations being involuntarily "trapped," or, without sufficient capability or resources to move away[16]. Observing immobility would be in line with prior studies of comparable sudden-onset hazards[11,12,17], and would correspond with the expectation of housing affordability constraints on out-migration[22] as well as environmental amenity pulls to remain in fire-affected places[8,20]. The null hypothesis, therefore, is that no change in out- or in-migration will be observed in wildfire-affected areas relative to neighboring, unaffected areas.

In this work, we analyze the migration effects of the top 10% most destructive wildfires in the contiguous U.S. ($N = 519$) between 1999 and 2020. We construct a temporally and spatially harmonized dataset that combines data on wildfire-related structure loss with data on migration at the census tract scale, which are the most comparable spatial unit to neighborhoods[36]. The structure loss data are from the U.S. National Incident Command System/Incident Status Summary Forms (hereafter "ICS")[33], linked to two sets of wildfire spatial burn footprints[37,38]. Together, these data offer one of the most comprehensive and detailed data sources of wildfires and their impacts within the U.S. The migration data are based on the Federal Reserve Bank of New York/Equifax Consumer Credit Panel (CCP) and estimate the number of credit-visible residents whose address changes tracts between two adjacent quarters. The CCP is an anonymous random sample from the Equifax credit files which can be used to calculate quarterly estimates of the probability of in-migration to and out-migration from census tracts[39,40]. These data allow us to investigate the effects of wildfire destruction on human migration, stratified by level of fire severity.

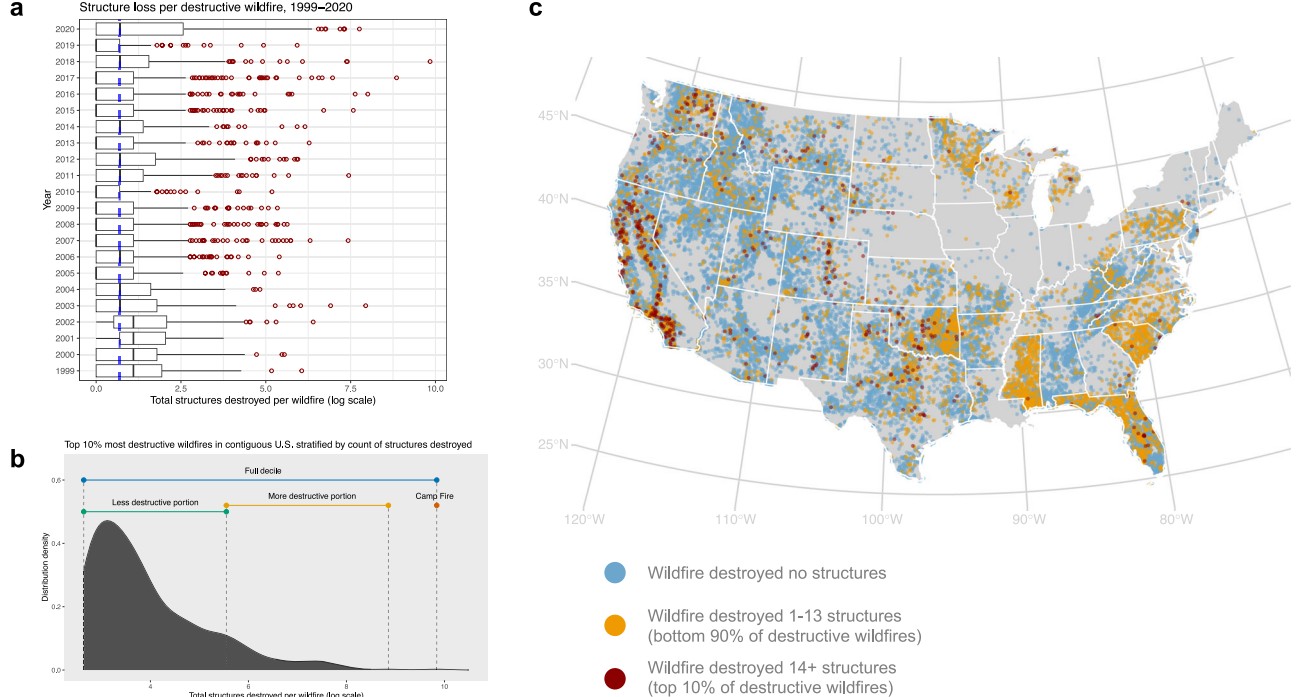

**Fig. 1 | A relatively small proportion of wildfires cause widespread structure loss. a** Boxplots show the distribution of annual structure damage per destructive wildfire among all wildfires reported by the ICS dataset in the U.S. between 1999 and 2020 that destroyed 1 or more structures (*N* = 5406). The left whisker indicates the minimum value to the 25th percentile, the right whisker indicates the 75th percentile to the maximum value, the left side of the box indicates the 25th percentile, the right side of the box indicates the 75th percentile, and the line within the box indicates the median. Red dots indicate extreme events that destroyed more structures than maximum values. The dotted blue line indicates a global median of structure damage across all years (2 structures destroyed). While the majority of destructive wildfires affected a relatively small number of structures (90% impacted fewer than 14), a small number of events had an outsized contribution to the total number of structures destroyed. **b** Figure shows the probability distribution of structures destroyed per wildfire event among the top decile of most destructive wildfires that include spatial details the ICS dataset in the contiguous U.S. from 1999 to 2020 (*N* = 529). Within this top decile of wildfires (those that destroyed

between 14 to 18,804 structures), the count of structures destroyed per event is highly right skewed. The figure shows how we stratified events for subsequent analysis into the less destructive portion of the decile distribution (green line), more destructive portion of the decile distribution (gold line), and the single most destructive event in the distribution, the Camp Fire (red point). We also analyzed the full decile of events (blue line). **c** Map shows the geographic distribution of wildfires with destruction levels and points of origin reported in the ICS dataset from 1999 to 2020 in the contiguous U.S. (*N* = 32,296). Each point on the map represents a wildfire point of origin, where the color indicates level of structure loss caused by the fire. Blue dots indicate fires that caused no structure loss; yellow dots indicate the majority of destructive wildfires (90%) that destroyed 1–13 structures; and red dots indicate the most destructive wildfires (top 10%), which destroyed between 14 and 18,804 structures. We focused our analysis on the latter group, analyzing only the most destructive wildfires. Sources: Wildfire data are from the U.S. National Incident Management System/Incident Command System[33] and state boundaries are from the U.S. Census Bureau.

## Results

### Extreme, outlying wildfire events drive the majority of structure loss

The majority of wildfires ignited between 1999 and 2020 caused no damage to the built environment (*N* = 29,216, 84.4% of all incidents), with a relatively small proportion destroying one or more structures (*N* = 5406, 15.6%) (Fig. 1c). Within this subset of "destructive wildfires," levels of destruction were non-linear across events; a small number of wildfires destroyed a disproportionately large portion of all structures (Fig. 1a). During the period examined, the top ten most destructive fires caused 39.5% of all wildfire-related structure loss, and the single largest event, the 2018 Camp Fire, was responsible for 17.2% of all wildfire structure loss over more than two decades. We focus our analysis on the top decile of most destructive wildfires, further stratifying this subset of events by severity (Fig. 1b).

### Wildfire structure loss drives increased out-migration only at highest severity levels

Our results indicate that, in the rare cases in which wildfires influenced migration, they did so through our first hypothesized pathway: direct impacts to the built environment (Table 1, Fig. 2). Wildfires were only associated with heightened out-migration in tracts that experienced

the highest levels of structure loss, indicating that wildfire effects on migration were non-linear and only observed beyond a certain destruction severity threshold. Furthermore, migratory effects were primarily constrained to the first year following the event, and, in most cases, did not extend beyond this initial time period.

Our analysis presents results for the full decile of most destructive wildfires (*N* = 519), as well as for three subsets of these events stratified by destruction severity: the less destructive portion of the decile (*N* = 463), the more destructive portion of the decile (*N* = 55), and the most destructive event, the Camp Fire (*N* = 1). Each subset of wildfires is presented in its own row in Table 1. To address the potential for spatial spillover and ensure the robustness of our findings, we report regression coefficients derived from comparison to three distinct sets of control groups. Each set of controls was selected from a different distance away from the burned tracts (0–5 miles, 5–25 miles, and 25–50 miles, shown in Fig. 2c) to reflect both heterogeneity in and uncertainty about the extent of spatial spillover.

When analyzing the full top decile of destructive wildfires (between 14 and 18,804 structures destroyed per event), we observed significant and positive out-migration effects during the event quarter when using the 5-mile and 50-mile control sets (Table 1, columns A and C). This migratory effect became larger in magnitude during the first

**Table 1 | Coefficients for the effects of wildfire destruction on migration probability**

| Wildfire subset | Time period | Out-migration probability | | | In-migration probability | | |
|---|---|---|---|---|---|---|---|
| | | (A) 0–5-mile controls | (B) 5–25-mile controls | (C) 25–50-mile controls | (D) 0–5-mile controls | (E) 5–25-mile controls | (F) 25–50-mile controls |
| Full decile of most destructive wildfires (14–18,804 structures destroyed, n = 519 events) | Event quarter | 0.0034* (0.0015) p = 0.0238 | 0.0018 (0.0014) p = 0.2084 | 0.0034* (0.0014) p = 0.0148 | −0.0004 (0.0017) p = 0.8209 | −0.0006 (0.0016) p = 0.7025 | −0.0001 (0.0016) p = 0.9643 |
| | First year post-event | 0.0048*** (0.0010) p = 0.0000 | 0.0045*** (0.0010) p = 0.0000 | 0.0044*** (0.0010) p = 0.0000 | 0.0023 (0.0016) p = 0.1519 | 0.0023 (0.0013) p = 0.0819 | 0.0020 (0.0014) p = 0.1512 |
| | Second year post-event | 0.0004 (0.0012) p = 0.7158 | 0.0012 (0.0009) p = 0.1596 | 0.0011 (0.0008) p = 0.1828 | 0.0009 (0.0013) p = 0.4738 | 0.0010 (0.0014) p = 0.4796 | 0.0010 (0.0014) p = 0.4552 |
| Less destructive portion of wildfire distribution (14–257 structures destroyed, N = 463 events) | Event quarter | −0.0015 (0.0022) p = 0.4908 | −0.0044* (0.0021) p = 0.0317 | −0.0019 (0.0018) p = 0.2924 | −0.0013 (0.0022) p = 0.5585 | −0.0024 (0.0023) p = 0.2877 | −0.0004 (0.0020) p = 0.8596 |
| | First year post-event | 0.0003 (0.0013) p = 0.8288 | 0.0014 (0.0013) p = 0.3012 | 0.0002 (0.0012) p = 0.8700 | −0.0006 (0.0021) p = 0.7607 | 0.0013 (0.0022) p = 0.5686 | −0.0010 (0.0020) p = 0.6321 |
| | Second year post-event | 0.0004 (0.0017) p = 0.7861 | 0.0022 (0.0014) p = 0.1245 | −0.0001 (0.0016) p = 0.9513 | −0.0013 (0.0029) p = 0.6492 | −0.0000 (0.0030) p = 0.9984 | −0.0001 (0.0025) p = 0.9632 |
| More destructive portion of wildfire distribution (258–7010 structures destroyed, N = 55 events) | Event quarter | 0.0030 (0.0022) p = 0.1727 | 0.0031 (0.0020) p = 0.1320 | 0.0041* (0.0019) p = 0.0324 | −0.0010 (0.0026) p = 0.7038 | 0.0016 (0.0021) p = 0.4452 | −0.0011 (0.0020) p = 0.5677 |
| | First year post-event | 0.0047*** (0.0013) p = 0.0004 | 0.0026 (0.0014) p = 0.0684 | 0.0041*** (0.0010) p = 0.0001 | 0.0024 (0.0021) p = 0.2447 | −0.0004 (0.0015) p = 0.7806 | 0.0002 (0.0015) p = 0.8747 |
| | Second year post-event | 0.0014 (0.0011) p = 0.2030 | −0.0006 (0.0014) p = 0.6473 | 0.0006 (0.0009) p = 0.5149 | 0.0016 (0.0018) p = 0.3889 | 0.0016 (0.0016) p = 0.3107 | −0.0002 (0.0012) p = 0.8708 |
| Most destructive event in wildfire distribution, 2018 Camp Fire (18,804 structures destroyed, N = 1 event) | Event quarter | 0.0535* (0.0188) p = 0.0065 | 0.0693*** (0.0190) p = 0.0007 | 0.0694*** (0.0179) p = 0.0003 | 0.0060 (0.0089) p = 0.4998 | 0.0060 (0.0079) p = 0.4527 | 0.0295** (0.0097) p = 0.0038 |
| | First year post-event | 0.0680*** (0.0191) p = 0.0009 | 0.0833*** (0.0208) p = 0.0003 | 0.0828*** (0.0182) p = 0.0000 | 0.0131 (0.0074) p = 0.0830 | 0.0169* (0.0082) p = 0.0459 | 0.0179** (0.0065) p = 0.0078 |
| | Second year post-event | 0.0191* (0.0088) p = 0.0353 | 0.0258* (0.0098) p = 0.0117 | 0.0162 (0.0081) p = 0.0519 | 0.0103* (0.0049) p = 0.0405 | 0.0119* (0.0049) p = 0.0197 | 0.0060 (0.0052) p = 0.2549 |

***p < 0.0021; **p < 0.01; *p < 0.05. *** indicates Bonferroni-adjusted p value threshold. Table reports the interaction terms (Temporal Period*Treatment) from difference-in-differences models, which use weights derived from coarsened exact matching. Two-sided p-values are reported and rounded to the fourth decimal place. Robust standard errors are clustered at the census tract level and shown in parentheses. Full regression results reported in Supplementary Information Tables S.I.1–S.I.4. Sources: Federal Reserve Bank of New York/Equifax Consumer Credit Panel and U.S. National Incident Management System/Incident Command System[33].

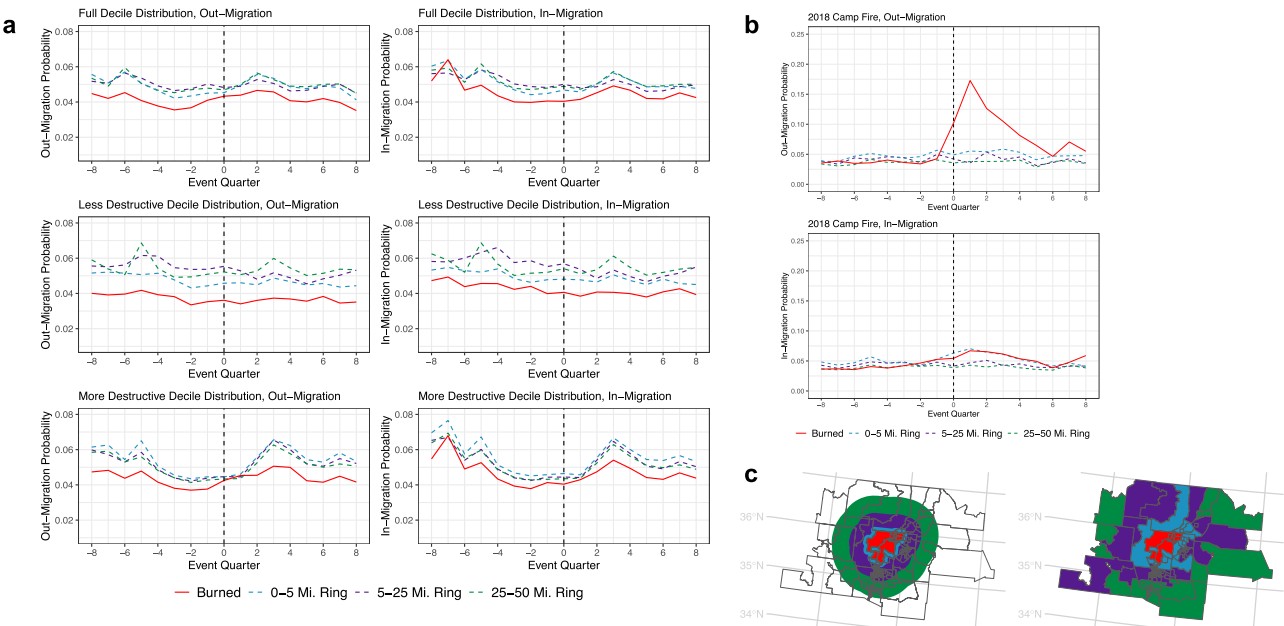

**Fig. 2 | Out-migration effects of wildfire structure loss are observed only following the most destructive events. a** Figures show evolving, unweighted out-migration probabilities (left) and in-migration probabilities (right) among three subsets of destructive wildfires: (1) full top decile distribution (14–18,804 structures destroyed, N = 519 wildfires), (2) less destructive portion of the top decile (14–257 structures destroyed, $N$ = 463 wildfires), and (3) more destructive portion of the top decile (258–7010 structures destroyed, N = 55 wildfires). Control tract migration probabilities are shown in blue, purple, and green. Vertical dashed line indicates the quarter in which the wildfire occurred. **b** Figures show evolving, unweighted out-migration probabilities (left) and in-migration probabilities (right) before and after the 2018 Camp Fire. Control tract migration probabilities are shown in blue, purple, and green. Vertical dashed line indicates the quarter in which the wildfire occurred. **c** For each wildfire event, we selected three rings of control tracts for each cluster of burned tracts (shown in red). Figure shows control selection for the 2000 Cerro Grande Fire in New Mexico. The buffer from the outer edge of burned tracts to 5 miles away is shown in blue; the buffer from 5 to 25 miles is shown in purple; and the buffer between 25 and 50 miles away from the edge of treated tracts is shown in green (left). These buffers are then intersected with spatially overlapping tracts (right). Sources: Migration data are from the Federal Reserve Bank of New York/Equifax Consumer Credit Panel, wildfire data are from the U.S. National Incident Management System/Incident Command System[33], and tract boundaries are from the U.S. Census Bureau.

year after the event, with estimates ranging from 0.0048 when using the 5-mile control set and 0.0044 when using the 50-mile control set (Table 1, columns A-C). Put differently, burned tracts experienced 4–5 additional movers per thousand residents, on average, in the year after the fire compared to unburned tracts.

We subsequently analyzed different components of the full decile and observed that wildfires in the less destructive portion of the decile (between 14 and 257 structures destroyed) caused almost no significant changes to out-migration probability. There was only a slight decrease in out-migration probability in the event quarter when using the 25-mile control set (Table 1, column B), however this effect was not evident when using either alternative control set. The lack of migratory effects among events with lower levels of structure loss and during any disaster or post-disaster time period means that, absent high levels of structure loss, we did not observe population-level migration changes that would indicate wildfires spurred changing residential preferences or capabilities and, subsequently, migration decisions.

When we next examined wildfires in the more destructive portion of the decile (between 258 and 7010 structures destroyed), a migratory effect associated with structure loss was clearly evident. Among the more destructive portion of the top decile, there were four additional out-migrants per thousand residents, on average, during the event quarter, however this effect was only observed when using the 50-mile control set (Table 1, column C). The effect was more pronounced in the first year following the event, where we observed five and four additional out-migrants per thousand residents when using the 5-mile and 50-mile control sets respectively (Table 1, columns A and C). We did not observe any significant differences in out-migration the second year following the event, indicating that, during this period, migration

trends returned to a similar trajectory as their neighboring control tracts.

Turning to the fourth subset, which includes the single most destructive fire, we saw that the out-migration effect of the Camp Fire (18,804 structures destroyed) was larger in magnitude and longer in temporal duration than any other subset of destructive wildfires. This suggests that both migration driven directly by structure loss as well as indirect wildfire-related migration both occurred. During the event quarter, models indicate that burned tracts experienced between fifty-three and sixty-nine additional out-migrants per thousand residents compared to unburned control tracts (Table 1, columns A-C). This substantial increase in out-migration immediately following the event indicates that the large scale of the Camp Fire's destruction led to initial displacement through structure loss.

Following the event period, the migratory effect grew in magnitude during the first post-fire year, where burned tracts experienced between 68 and 83 additional out-migrants per thousand residents per quarter compared to unburned control tracts. This translates to a more than threefold increase in the magnitude of out-migration probability among burned tracts from the two years prior to the fire to the first year following the event quarter. Compared to the more destructive portion of the top decile (between 258 and 7010 structures destroyed), the migratory effect of the Camp Fire during the first year after the event was between fourteen and twenty times as large. Unlike any other subset of destructive wildfires, models indicate that the Camp Fire's out-migration effect was still significant in the second year after the event. Burned tracts experienced between nineteen and 26 additional out-migrants per thousand residents when using the 5- and 25-mile control sets respectively (Table 1, column A and C). This

elevated out-migration trend in the two full years following the Camp Fire provides evidence supporting our hypothesis of indirect wildfire effects on migration, which we theorize are driven by changing residential preferences and capabilities, rather than destruction of the built environment. After the initial spike in out-migration driven by rapid structure loss, residents continued to leave the area.

**Wildfire structure loss has minimal impact on in-migration trends**

Finally, we examined trends in in-migration, hypothesizing that indirect effects of wildfires will result in reduced in-migration during and after the event period, as potential in-migrants avoid fire-affected places. Across the full top decile, less destructive, and more destructive portions of the top decile, there were no significant differences in post-fire in-migration among burned tracts relative to any set of control tracts (Table 1, columns D–F). It was only following the Camp Fire that we observed a significant increase in in-migration probability, starting during the event quarter, where there were an additional 30 in-migrants per thousand residents relative to the 50-mile control set (Table 1, column F). This positive effect on in-migration continued during the first year when using both the 25-mile and 50-mile control sets, and again during the second year, when using the 5- and 25-mile control sets (Table 1, columns D–F). We interpret this increase in in-migration as evidence of what is known as "recovery migration," wherein returning and new residents arrive in a disaster-affected area following an initial displacement event[41,42].

When examining parallel trend plots for Camp Fire in-migration (Fig. 2b), we observed some evidence of spatial spillovers in the nearest set of control tracts, those between zero and five miles from burned tracts. As the red line indicating mean in-migration probability in burned tracts rises and remains elevated during the event and post-event quarters, so too does the mean in-migration probability for the 5-mile ring, which is shown in blue. The two trends evolve along very similar trajectories, whereas in-migration among 25-mile and 50-mile control tracts remains relatively flat in the post-year period. This spatial spillover is reflected in the non-significance of coefficients for the 5-mile ring comparison in the event and post-event year interaction terms (Table 1). Given how large the effect of the Camp Fire was on out-migration, it is possible that this in-migration spillover reflects residents leaving the immediately burned area and moving into nearby tracts.

## Discussion

Despite the robust growth of climate migration research over the past decade[11,12], wildfires remain understudied in this field[13]. Existing research on the effects of comparable sudden-onset hazards indicates that a spectrum of different migratory responses are possible. On one hand, many past studies have shown that such events result in relative immobility[11,12,17]. However, on the other hand, studies focused on extremely destructive hurricanes and tsunamis have documented heightened out-migration and subsequent recovery migration[23–26,41,42]. Our analysis of wildfires across a range of destruction levels reflects this heterogeneity of effects observed in prior literature, illustrating the prevalence of severity thresholds at which wildfires influence migration in the U.S. We show that immobility was the most common response to destructive wildfires, however, for the smaller number of highly destructive fires, we observed increased out-migration.

Our study draws on wildfire data that document exact wildfire structure loss counts[33], allowing us to stratify our analysis by severity level and to subsequently test for different types of wildfire-related migration. We paired these data with migration estimates from the Federal Reserve Bank of New York/Equifax Consumer Credit Panel, which has been minimally used for migration research but offers improved spatial resolution over traditional migration data. Together, these data sources make possible analysis at the census tract scale,

which approximates neighborhoods, offering a level of spatial granularity that has not been previously possible in most multi-decadal environmental migration studies. By analyzing a large number of events, our analysis further provides generalizable findings on an understudied hazard within environmental migration scholarship.

We investigated the 519 most destructive wildfires in the contiguous U.S. between 1999 and 2020, examining direct and indirect pathways of wildfire-driven impacts on human migration. We first tested for migration effects through direct damage to the built environment, wherein heightened out-migration occurs following high levels of structure loss. Second, we examined whether wildfires influenced migration indirectly, through mechanisms apart from structure loss. Through this pathway, residential preferences to remain in place or migrate as well as residents' capabilities to realize such preferences may change as a result of a fire, in turn affecting population-level migration trends.

Our findings support our first hypothesis, that wildfires affect migration patterns non-linearly at high levels of structure loss, as housing and other infrastructure are destroyed, and residents subsequently relocate. We found that only a small portion of destructive wildfires caused a migratory response, and such rare events influenced mobility primarily through destruction to the built environment. Even among the top ten percent of the most destructive wildfires in the contiguous U.S., it was only the most extreme among these events that caused an increase in out-migration. This was reflected in significant wildfire effects on out-migration among wildfires in the most destructive portion of the top decile (258–7010 structures destroyed), and the largest magnitude of out-migration effects observed after the single most destructive event, the 2018 Camp Fire. Migration following highly destructive events is in keeping with prior research that has documented direct displacement following extreme sudden-onset disasters[23–26] and non-linear relationships between migration and hazard severity[17,34]. It is also in line with emerging literature on wildfire-related mobility, which has documented temporary population displacement following two highly destructive events, the Mendocino Complex and Woolsey Fires in California[30]. However, our research design ultimately does not allow us to distinguish between residents moving away because their own dwellings were destroyed, because their local environment experienced high levels of destruction, or a combination of both. These possible pathways should be investigated in future research with qualitative methods focused on migration decision-making.

We further hypothesized that, separate from direct destruction to structures, wildfire impacts on the biophysical, economic, and social dynamics of a place would influence residents' desires to remain living there and/or their ability to do so. However, in most cases, we did not find evidence indicating that wildfires influenced migration patterns through this indirect pathway via residents' changing mobility preferences or capabilities. While wildfires can influence human migration at high levels of structure loss, the majority of wildfire events between 1999 and 2020 did not reach this destruction threshold and, subsequently, did not result in changes to existing migration trends. Following the majority of destructive wildfires in this study (14–257 structures destroyed, 89% of the 519 wildfires examined), we observed no significant increase in out-migration, indicating that immobility is a common response to wildfire, as it is among other hazards[11,12,14–17]. Furthermore, the rare spikes in out-migration following the most destructive events were almost all temporally constrained to the disaster period and first year following the event, and did not remain elevated in the second post-event year. The only exception to this trend was following the Camp Fire, in which out-migration from burned tracts remained elevated for the entire temporal period examined. Finally, we observed no declines in in-migration following wildfire events. Rather than being deterred by substantial wildfire destruction, in-migrants arrived in fire-affected tracts at the same rate

that they did prior to the fire and relative to neighboring, unburned tracts. Together, these findings suggest that, during the study period, wildfires that did not cause very high levels of structure loss also did not influence residential mobility preferences and/or capabilities sufficiently to affect population-level migration trends.

Prior environmental migration scholarship conducted across hazard types finds broad variability in the direction and magnitude of migration response. We showed that, further, the migration response to the same hazard can vary widely across severity levels, increasing non-linearly at the highest level of impact. Our findings speak to the outsized effects of the most extreme environmental events on human migration. Fires are a common environmental phenomenon occurring across many parts of the U.S. (Fig. 1); it is only a much smaller subset of rare, but extremely destructive wildfires that have directly impacted migration through structure loss. This finding is important for situating a general understanding of wildfire-related mobility in the contiguous U.S. – namely, that immobility is the most common response to destructive wildfires. Climate mobility scholars have recently begun emphasizing such findings that have historically been treated as null results of lesser interest, arguing for the importance of studying immobility, especially in the context of intensifying environmental hazards[14,16,19]. Future research should investigate how both individual aspirations and macro-level structural conditions collectively inform the mobility of residents living in fire-prone places.

While we observed immobility as the most common response to destructive wildfires, we also know that the rate of wildfire-driven structure loss in the U.S. has been increasing over time[1], with a substantial number of outlying extreme events occurring in recent years (Fig. 1). Absent major adaptation efforts, if the recent intensification of wildfire destructiveness continues, our findings suggest that we may expect to observe more direct displacement caused by extreme wildfires in the future. Although we did not observe substantial evidence of indirect wildfire-mobility effects, in which residents began leaving or avoided moving into fire-affected regions absent major structure loss, these effects may yet emerge in the future as the wildfire regime continues to change. Future research should examine how these pathways of wildfire-related migration evolve. Additionally, research in this area could examine whether more recent extreme events and events outside of the contiguous U.S., such as the 2023 Maui Fire in Hawaii, have similar migration effects as those found in this analysis.

Our research design provides a number of important advances to the emerging study of wildfire-related mobility. First, because our wildfire data measure exact counts of structures destroyed at a fine spatial scale, we were able to stratify our analyses by level of wildfire severity. This is a distinct approach from previous wildfire migration studies, which have either investigated a very small selection of events[30–32], or have made minimal distinctions in event severity among many events[20]. By stratifying our analysis across levels of wildfire destruction, we are able to examine thresholds in wildfire-migration relationships, which is an important area of investigation given prior research suggesting non-linear migration responses to other environmental hazards[11,12,17,34]. Second, our data allow us to examine wildfire-related migration at the census tract scale, the spatial unit that most closely approximates neighborhoods[36]. This spatial scale is critical conceptually, given that prior migration research generally documents short-distance moves in response to environmental changes[15]. It is also technically important for the study of wildfires, because their area of direct impact tends to be small relative to the land area of counties, the coarser spatial unit used in prior studies most similar to ours[20,32] (see Supplementary Information 2 for a more detailed discussion of wildfires and spatial scale). Finally, compared to past studies, our quasi-experimental design comparing burned tracts to counterfactual unburned tracts offers improved causal identification of wildfire effects on migration. This approach has not previously been used to study wildfire-migration relationships and is especially important for

research on environmental hazard impacts, given the potential for confounding events[43]. Our use of three distinct sets of control groups further allows us to ensure the robustness of our findings and identify spatial spillovers outside of immediately burned regions. Together, these elements of our research design allow us to comprehensively identify nonlinear effects of wildfire destruction at a local scale.

Our study has several limitations that we anticipate can be addressed as future research continues to expand knowledge on wildfire-mobility dynamics. First, our study design did not identify residential moves within tracts. As a result, it is possible that wildfire destruction may cause changes to population mobility at a finer spatial scale than we were able to observe. Such a pattern would be in keeping with findings from a Colorado-based survey, in which residents in a fire-affected region who desired to move preferred nearby destinations[27]. However, even if such within-tract residential mobility were taking place, it would still affirm our study's broader conclusion: residents by and large did not migrate out of fire-prone areas after less destructive events. Additionally, our aggregated census tract-level approach is not able to distinguish between individual residents whose dwellings were located within a burned tract but not within the burn footprint, and those whose dwellings were located directly within the burn footprint. Because some wildfires fall within a census tract but do not burn that tract's entire area, our approach necessarily included some unexposed residents in the treated condition. This may mean that our results underestimate the magnitude of migratory effects.

A second limitation of our approach is that the CCP migration data generally cannot be demographically decomposed[40]. Using these data, we are limited in our ability to analyze potentially different migration trends across axes of difference such as race, ethnicity, or nativity. While our approach provides a broad picture, we cannot determine whether particular demographic groups are more or less likely to migrate in response to wildfire destruction. This limitation is not unique to the CCP migration data, and we are aware of no publicly available migration data source that has extensive spatial and temporal coverage, fine-grained spatial and temporal units, and demographic decomposability. Future work creating such data would greatly expand the scope of environmental migration research, enabling lines of inquiry focused more explicitly on disproportionate impacts and questions of equity.

Finally, it is important to note that the CCP migration data only include residents with a Social Security Number (SSN) and a credit history, and therefore under-represent relatively younger and financially disadvantaged people[44,45]. As such, the CCP sample is not necessarily representative of the full U.S. population in all places. This challenge is endemic to many commonly-used forms of migration data; for instance, the Internal Revenue Service's county-to-county migration data includes only residents who file taxes[46] and mobility data derived from mobile phones only sample from residents who use a cell phone[47]. There is a tradeoff to using migration data such as these that offer broad geographic and temporal coverage, but do not fully capture all subpopulations that may be especially vulnerable to wildfire impacts. For example, a case study of the Camp Fire found that the residential structure types most likely to house lower-income residents, mobile home residents, and renters had a higher probability of being destroyed in the fire, suggesting that these populations were more susceptible to housing loss due to characteristics of the built environment[48]. While we clearly detected a significant migratory effect from the Camp Fire, the CCP's underrepresentation of financially disadvantaged residents means that we may have underestimated the overall effect size for this particular event if these residents were not fully represented in the data. Similarly, in a case study of the 2017 Thomas Fire in California, researchers highlighted the ways that undocumented immigrants who worked in affected areas were both highly impacted by the fire but simultaneously not visible in official census statistics[49]. Focused attention on the experiences of vulnerable

subpopulations with wildfire is needed, and must be conducted with tailored data that can overcome limitations of existing national-level datasets. Yet, such analyses would need to address considerable privacy concerns that arise when studying demographically identified groups at small spatial scales.

The heightened out-migration observed after relatively rare but highly destructive wildfires invites further study focused more closely on patterns of in-migration in the years following the event. The concept of "recovery migration" adopted in scholarship on hurricanes encompasses both returning residents and new in-migrants[41,42], and others have further highlighted the temporary in-migration of individuals drawn by disaster cleanup employment[50]. This area of research is generally understudied relative to other aspects of environmental migration[10], and future research should analyze these distinct forms of in-migration after destructive wildfires. Existing studies suggest that several possible dynamics may be at play, including post-wildfire gentrification[48], as well as the continued push of residents into more affordable but also more fire-prone places[22].

In this study, we present a broad examination of wildfire's impacts on human migration patterns in the contiguous U.S., addressing a scarcity of wildfire-focused research in environmental migration scholarship[13]. Emerging scholarship on this topic to date has been geographically focused on North America, yet wildfires are a global phenomenon[5,6]. Prior research conducted in countries with substantial agricultural sectors generally finds more pronounced environmental migration effects, as environmental changes alter agricultural productivity, thereby influencing household income and subsequent migration[10,12,17]. This pattern suggests that, in different geographic contexts, wildfires may influence migration differently, with potentially stronger effects in agriculturally-dependent regions. Future research should investigate wildfire impacts on migration across the broad geography of fire-prone places, with special attention to the different causal pathways through which fire may influence mobility.

## Methods
### Data construction
We constructed a longitudinal dataset of wildfire destruction and quarterly out- and in-migration probabilities at the census tract scale. Wildfire destruction metrics were adapted from administrative records collected in the U.S. National Incident Management System/ Incident Command System, archived by the interagency National Wildfire Coordinating Group, and subsequently processed by St. Denis et al. ("ICS")[33]. The ICS records encompass all documented wildfires in the U.S. that require the establishment of an incident management team. Drawing on St. Denis et al.'s procedure to create a spatio-temporal version of the data, we used the ICS's linkage to wildfire perimeters from the Monitoring Trends in Burn Severity database[38] and the Fire Events Delineation (FIRED) database[37] to produce census tract- and quarter-level wildfire data based on 2010 tract boundaries. We selected census tracts as our unit of analysis because they approximate a measure of neighborhoods, generally including between 1200 to 8000 residents[36]. No single unit of analysis perfectly corresponded to the wide range of wildfire burn footprint sizes in our data. However, the granularity of census tracts is better-suited to match the spatial scale of burn footprints, which are generally much smaller than the next largest administrative unit—counties—which have been used in prior wildfire research (see Supplementary Information 2 for additional details on spatial unit selection)[20,32]. We obtained tract and state boundaries from the U.S. Census Bureau through the National Historical Geographic Information System (NHGIS) and R *tigris* package respectively[51].

The ICS dataset is one of the most comprehensive longitudinal sources of wildfire data for the U.S. For each wildfire event, the ICS reports the total number of structures destroyed, a measure that includes residential, commercial, outbuilding, and mixed-use structures. We utilize data from the full temporal period available in the most recent publication of the ICS, which covers 1999 through 2020. 1999 was the first year for which the National Wildfire Coordinating Group provided the raw data from which the ICS is produced. 2020 represents the most recent year through which the ICS has been cleaned[33].

A major benefit of the ICS dataset is that it reports direct measures of hazard impact rather than the dollar value of damaged property. The latter approach to disaster data reporting, while commonly used, is unable to distinguish between the destruction of a small number of high-value structures and a high number of low-value structures. The conflation of number of structures damaged or destroyed with the estimated monetary value of damages to structures distorts damage estimates, overstating damages in areas with high property values and understating damages in areas with low property values. The ICS counts of destroyed or damaged structures thus provide a more direct measure of hazard impact[33]. However, it does not account for wildfire impacts on wildlands, agricultural lands, or livestock, which could potentially influence migration via impacts on environment-dependent livelihoods such as forestry, farming, or environmental amenity-based tourism.

Migration measures come from the Federal Reserve Bank of New York/Equifax Consumer Credit Panel (CCP). The CCP is an anonymous five percent random sample drawn from the credit histories maintained by Equifax. It contains panel data on over 10 million individuals. The consumer credit histories are built from the monthly reports Equifax receives from mortgage lenders, credit card issuers, student loan servicers, and other debt holders. Equifax uses an algorithm to identify each individual's most likely current address from the addresses reported by all of a borrower's creditors. Equifax provides the census tract containing the selected address in the CCP data. The street addresses themselves are withheld for anonymity, as are all names and Social Security Numbers. A unique anonymous identifier is assigned to each borrower, allowing researchers to build individual-level quarterly histories[40]. To account for differences in population size between tracts, we used the proportion of individuals in a tract who moved into or out of the tract as the dependent variable for our analysis. Unfortunately, the CCP does not contain demographic information on the borrowers, such as sex, race, ethnicity, or nativity.

The Federal Reserve Bank of New York/Equifax Consumer Credit Panel (CCP) has several advantages over other sources of data on residential migration. Compared to U.S. Census Bureau surveys that measure migration, such as the American Community Survey or the Current Population Survey, the CCP's large sample size provides statistical power necessary for analyses at smaller spatial and temporal scales[45]. Compared to the widely-used Internal Revenue Service's county-to-county migration estimates (IRS), which report total counts of migrants between county pairs, the CCP provides individual-level records which report residential locations quarterly, as opposed to annually, strengthening temporal inference. Further, the individual-level records can be aggregated to fit a spatial unit, such as a state, county, or census tract. The finer temporal and spatial scales that are possible with the CCP make it highly attractive for the study of environmental shocks and migration responses[32].

The CCP also has several limitations. The data represent only those U.S. adults who have a Social Security Number (SSN) and a credit history. Therefore, coverage excludes the estimated 10–11% of adults who do not have a formal credit history and those without an SSN[44]. This means that younger and financially disadvantaged people are under-represented in the data[45]. As mentioned above, the CCP does not contain demographic information on the borrowers, such as sex, race, ethnicity, or nativity. These limitations mean that the dataset cannot be used to examine individual-level sociodemographic disparities in hazard-related migration.

Finally, to conduct our matching procedure, we processed tract-level landscape and population variables associated with wildfire risk[52,53]. These include elevation and slope derived from NASA's 90 m SRTM digital elevation map[54], and the percent of land in each tract belonging to specific land cover classes, derived from the 2019 National Land Cover Database[55]. Of available land cover classes, we utilized the percentage of a tract covered by forest, shrub/scrub, and developed land, which are associated with flammability[53]. We processed the variables above using Google Earth Engine's cloud computing platform[56]. In addition to these landscape characteristics, we also included a tract's total land area (where smaller indicates a more urbanized tract) and 2010 county-level population estimates[57].

## Stratification of wildfires by severity

Given the right skew of the wildfire destruction distribution, we chose to analyze only the most destructive decile of wildfires that destroyed structures (hereafter, "top decile"). The top decile encompasses events ranging from fourteen structures destroyed at the least destructive to 18,804 structures destroyed at the most destructive. However, even this most destructive top decile itself is right skewed, with the majority of events causing a lower level of destruction. For this reason, we subsequently stratified the top decile into four sets for analysis: (1) the full decile distribution ($n = 519$), (2) the less destructive portion of the decile distribution ($n = 463$), (3) the more destructive portion of the decile distribution excluding the Camp Fire ($n = 55$), and (4) the most destructive event in the decile distribution, the Camp Fire ($n = 1$) (shown in Fig. 1b). We subdivided the top decile into these groups with the aid of Jenks natural breaks classification, which is a data classification method that minimizes variation within groups[58].

## Analytical strategy

We used a difference-in-differences (DID) strategy to model out-migration and in-migration probabilities in wildfire "treated" tracts (e.g., tracts containing the burn footprint) comparing them to unburned "control" tracts. We compared migration probabilities during the eight quarters preceding the event with the event quarter and eight quarters after the event. A separate DID model was fitted for each subset of events and control rings (see section 2.4). The model takes the form:

$$mp_{it} = \beta_0 + \beta_1 \text{Treat}_i + \sum_{t=0}^{T} \beta_{2t} \text{Period}_t + \sum_{t=0}^{T} \beta_{3t} \left( \text{Treat}_i * \text{Period}_t \right) + \varepsilon_{it}$$

Where $mp_{it}$ is a measure of migration probability in tract $i$ in time period $t$, which is defined as the total number of movers into or out of a tract divided by the total population of the tract at the start of the period. $\text{Treat}_i$ indicates whether a tract was burned in a wildfire event ("1") or an unburned control tract ("0"). $\text{Period}_t$ indicates whether the time period was pre-fire ("0" is the reference category), the event quarter ("1"), the first year after the event quarter ("2"), or the second year after the event quarter ("3"). We modeled multiple post-event temporal periods rather than a binary post-fire period to test whether migratory effects differed over time. The interaction terms between $\text{Treat}_i$ and each of the three event and post event $\text{Period}_t$ terms are the primary DID coefficients of interest. They reflect whether the change in migration probability between the pre-fire period and subsequent time periods was significantly different between burned and unburned tracts. $\varepsilon_{it}$ represents residual errors. We report robust standard errors clustered at the tract level and include a Bonferroni-adjusted $p$ value threshold of 0.0021. For ease of interpretation, we transformed the interaction coefficients to report X number of migrants per ten-thousand residents. We conducted analyses using the *estimatr* package[59] in R statistical software versions 4.3.3 and 4.4.0 and reported two-sided $p$ values.

Applied research analyzing longitudinal data has often used fixed effects (FE) to address concerns about omitted variables. However, recent methodological research suggests that this approach is inappropriate for certain causal research questions, and does not yield readily-interpreted, nonparametric causal estimators[60–62]. For this reason, we did not include fixed effects in our models, and instead addressed potential omitted variables bias through a matching procedure (described below). Matching treatment and control groups based on observed covariate values has recently been advanced as an alternative to FE models[63]. We nevertheless conducted additional sensitivity tests comparing our primary non-FE models to those with tract FE, quarter FE, and two-way FE. We performed these tests on the upper decile of wildfires using 0–5 mile controls, and found no substantive differences in the direction, magnitude, or statistical significance of the DID estimates. Additionally, we ran the same set of models but with a single pre- and single post-fire period, rather than three event and post-event periods. Here, we found that model coefficients followed the same patterns as our primary specification models, with significant increases in out-migration and no significant changes in in-migration in the post-fire period. These tests indicate that our findings are robust across a range of alternative specifications.

## Control selection

We matched control tracts to each treated tract through a two-step procedure. First, for each burned tract or cluster of tracts that correspond to a single wildfire, we calculated three rings of distance-based neighboring tracts. We did so by drawing buffers from the outer edge of burned tracts to 5 miles ("5-mile ring"), from 5 to 25 miles ("25-mile ring"), and from 25 to 50 miles ("50-mile ring") (Fig. 2c, left). We then intersected these buffers with spatially overlapping tracts (Fig. 2c, right) to create the final control tract selection for a given wildfire.

We selected three distinct sets of controls to address the potential for spatial spillover, in which the effects of a destructive wildfire travel beyond the immediate area in which the incident occurred. A recent study of Australia's Black Summer fires suggests that such spillovers can occur up to 5 km away from a directly burned area[43]. Because there is not sufficient empirical research from which to establish whether such spillovers are common across different wildfire events, we conducted analyses for each wildfire subset three times, each with a different ring of control tracts. Building these sensitivity tests into our analysis allowed us to rule out spatial spillovers for most wildfire subsets, and to identify a modest spatial spillover in the case of in-migration following the Camp Fire.

In cases in which a control tract also experienced a destructive wildfire within the seventeen-quarter observation window, the tract was removed from consideration as a control. This step ensured that treated units were not compared to control units that themselves were treated within the observation period. If a tract quarter was defined as a control for multiple fire-affected tracts, it was only counted once within a given pooled model. In- and out-migration probabilities vary more widely in tracts with small populations, which is in part due to the data's small sample size within these tracts. To minimize the influence of these outliers, observations with an in-migration probability greater than two standard deviations above the full dataset's mean in-migration were removed and observations with an out-migration probability greater than the maximum quarterly out-migration observed following the 2018 Camp Fire were removed.

After selecting three rings of potential control tracts for each wildfire, we next conducted coarsened exact matching (CEM) in order to balance covariates between treatment and control groups[64]. CEM has been used in prior disaster research to strengthen causal inference in quasi-experimental research designs[65]. We matched treatment and control tracts using a set of covariates that we selected based on their expected association with the treatment condition (experiencing a

destructive wildfire)[66]. Matching was conducted separately within each subset of wildfire events using the *MatchIt* package in R[67]. While ICS wildfire data are available for Hawaii and Alaska, certain covariates did not include coverage in these states. We therefore constrained our analysis to the contiguous U.S.

To evaluate covariate balance before and after matching, we examined the standardized mean differences between treatment and control groups of each covariate (Supplementary Information Figs. S.I.3–S.I.6). After matching, standardized mean differences were nearly all at or below 0.1, which is a threshold at which covariates are considered to be well-balanced. The Camp Fire model was the primary exception, where we used a smaller selection of covariates (tract size and percent developed, forest, and shrub or scrub) and matched covariates were better-balanced but did not all fall below the preferred 0.1 standardized mean differences threshold. These limitations were due to the smaller set of treated and control groups available for analyzing a single event, in contrast to the much larger N available for aggregated event analyses. Overall, results suggested that CEM substantially improved covariate balance across treated and control groups, with minimal reduction in the number of observations used for analysis (usually <10–15%).

### Reporting summary
Further information on research design is available in the Nature Portfolio Reporting Summary linked to this article.

## Data availability
The wildfire data and covariates used for coarsened exact matching are publicly available at refs. 33,54,55,57. Source data for figures are provided with this paper. The raw migration data from the Federal Reserve Bank of New York/Equifax Consumer Credit Panel (CCP) are available under restricted access to Federal Reserve System employees and cannot be shared due to Data Use Agreement terms. Source data are provided with this paper.

## Code availability
Codes developed to process the publicly available data listed above are available through OSF at https://osf.io/xa39e/.

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

## Acknowledgements

This work was supported by the U.S. National Science Foundation grant numbers 2001261 (K.M.), 2117405 (E.F., J.D., K.J.C.), and 1850871 (J.D., E.F., K.J.C.). K.M. and E.F. were supported by the Population Studies and Training Center at Brown University through the Eunice Kennedy Shriver National Institute of Child Health and Human Development (P2C HD041020). K.J.C. was supported by the Center for Demography and Ecology at the University of Wisconsin-Madison through the Eunice Kennedy Shriver National Institute of Child Health and Human Development (P2C HD047873) and the Wisconsin Agricultural Experiment Station. Further funding was provided by Earth Lab through the University of Colorado Boulder's Grand Challenge Initiative and the Cooperative Institute for Research in Environmental Science (CIRES). The views expressed in this report are those of the authors and are not necessarily those of the Federal Reserve Bank of Cleveland, the Board of Governors of the Federal Reserve System, Equifax, the NSF, or the NIH. Thank you to Justin Farrell, Emily Sellars, and Karen Seto for feedback on this research.

## Author contributions

K.M., E.F., J.D., and S.W. conceptualized and designed the study. K.M., S.W., and L.S. curated data. K.M. and S.W. wrote software and performed formal analysis. K.M. and E.F. wrote the original draft, and K.M., E.F., J.D., S.W., K.J.C., J.B., and K.P. contributed to review and editing of manuscript drafts. K.M. created visualizations. K.M., E.F., and J.D. supported funding acquisition.

## Competing interests

The authors declare no competing interests.
