## [Peer Review File · Nature Communications]

Rare and highly destructive wildfires drive human migration in the U.S.REVIEWER COMMENTS

Reviewer #1 (Remarks to the Author):

Thank you for the opportunity to review the article, "Rare, highly destructive wildfires drive human migration in the U.S.". This is a solid piece of research that has identified an important gap in the literature, created a clean research design and executed it, and has provided insightful results that drive the overall literature forward. I fully support publication with a few minor recommendations.

The authors have identified an important gap in literature. Very little is known, beyond anecdotal or regional evidence, about human mobility patterns and wildfire related housing losses—especially at larger scales. We know the frequency and severity of wildfire is on the rise in specific places, as are the number of people living in high fire risk regions, but we know little about whether people move or stay post-disaster. The authors propose and test hypotheses related to 1) out-migration directly related to housing loss, 2) outmigration indirectly related to other mechanisms and 3) a null hypothesis of no migration. Hypothesis 2 is tested mostly by observing migration patterns where too few buildings are destroyed to displace residents via housing loss, documenting out migration (or not) in several years post event, and in-migration patterns post event.

The quasi-experimental design is clever in its simplicity but should not be mistaken as an easy one to execute. The national database constructed between 1999 and 2020 is a novel and useful resource. I wonder about the start date of 1999 and why this date was chosen. A justification for both the start and end date would be helpful.

The findings are original and constructively inform and drive forward the conversation needed for how we respond to and recover from wildland fires. The dominant finding, and one that is a bit buried, is that immobility is the most common response to wildfire (which is similar to other hazards). I might encourage them to highlight this a bit more as a key takeaway. At the granular level, their data suggest that extreme, outlying wildfire events drive the majority of building destruction—the Camp Fire serving as the most destructive—and that wildfire building destruction drives increased out-migration only at the highest severity levels—again with the Camp Fire serving as the example. In short, they did not find population level migration changes. Finally, they provided support for hypothesis 2 with evidence that indirect wildfire effects had an impact on out migration two years after the event. I would caution the interpretation of results here. While the authors document the trend, they have no support about the motivations behind the migration (lines 212-213).

The authors analyze four subsets of the decile, but do not describe this approach at the outset. The description comes under the methods section (458-462). It would make it easier on the reader to provide some guidance about this approach on lines 158-159 before moving into the results description. In other words, a description like "our analysis proceeds in four stages and explores the deciles in the following manner..." Would make it easier for the reader to track.

The authors also count "structures" in the ICS dataset. Structures include both houses and outbuildings. While the ICS counts provide a direct measure of hazard impact, they do not separate houses from outbuildings. This may contribute to overstating impacts on houses if the figures are conflated. Likewise, a bit more careful attention to the language throughout and definitions up front would be useful. The terms buildings, houses and structures are used interchangeably, and greater precision focused on "structures" and what they are exactly is likely warranted.

Finally, the authors may have seen the recent publication by Radeloff et al. 2023 in Science on rising wildfire risk to houses in the US, especially in grasslands and shrublands. This is an informative and useful publication that also speaks to the trends in this article.

I really enjoyed reading this article and learned a lot from it. It was well written, well structured and I very much look forward to seeing it in print.

Toddi Steelman

Reviewer #2 (Remarks to the Author):

Thank you for the opportunity to review this paper, which is well-designed and well-written. The authors examine how wildfires have impacted human migration in the continental US between 1999 and 2020, focusing on the 10% most destructive wildfires in that period. They highlight two pathways for migration differences: direct (damage-driven) and indirect (changes in residential preferences and/or capabilities). I believe this to be a strong paper and recommend that the authors be given the chance to submit revisions toward future publication.

My primary comment is that I wish the article were more clearly rooted in the literature, which could help to contextualize and make sense of some of the current findings. It may be true that wildfires differ from other hazards in significant ways, in which case, the authors should explain why. Further, some discussion of broader migration trends for other reasons—such as people leaving fire-prone California because of cost-of-living—seems like important context here.

That said, the main finding of the paper—that people move away when their area experiences significant damage from wildfire—may be important, but it's not particularly surprising. It seems to me that others have addressed the issue of hazard (including wildfire) migration with greater attention to some explanatory factors, such as Winkler and Rouleau 2020 (<https://doi.org/10.1007/s11111-020-00364-4>). I think the authors need to make a better case for why this pretty intuitive finding is novel and/or important.

Is the implication that migration increases in highly impacted areas because more people lost their homes and left, or because there is a threshold of surrounding damage that people are less likely to tolerate, regardless of the survival of their home? The authors may be unable to answer this with their data, but the two possibilities speak to two very different processes.

Finally, there are some real equity concerns around this topic that are mostly ignored by the authors. I'd like to see this addressed more explicitly. The authors do briefly address it in the limitations, regarding problems with the data itself, but it is also relevant to the findings. For example, all in-migration is not the same; there is an important difference between people returning home after disaster (what the authors attribute to "recovery migration" after the Camp Fire), and others taking advantage of low property prices after a fire.

Again, the authors should be commended for a thoughtfully designed study and well-written paper. I believe with a little more work it can provide a valuable contribution to the existing literature on wildfire impacts and climate migration.

Response to reviewers

Thank you for the opportunity to revise and resubmit our manuscript; we very much appreciate the reviewers' close attention and constructive feedback. We have made a number of changes with a focus on deepening the paper's engagement with existing literature.

Reviewer 1

R1.1 Thank you for the opportunity to review the article, “Rare, highly destructive wildfires drive human migration in the U.S.”. This is a solid piece of research that has identified an important gap in the literature, created a clean research design and executed it, and has provided insightful results that drive the overall literature forward. I fully support publication with a few minor recommendations. The authors have identified an important gap in literature. Very little is known, beyond anecdotal or regional evidence, about human mobility patterns and wildfire related housing losses—especially at larger scales. We know the frequency and severity of wildfire is on the rise in specific places, as are the number of people living in high fire risk regions, but we know little about whether people move or stay post-disaster. The authors propose and test hypotheses related to 1) out-migration directly related to housing loss, 2) outmigration indirectly related to other mechanisms and 3) a null hypothesis of no migration. Hypothesis 2 is tested mostly by observing migration patterns where too few buildings are destroyed to displace residents via housing loss, documenting out migration (or not) in several years post event, and in-migration patterns post event. The quasi-experimental design is clever in its simplicity but should not be mistaken as an easy one to execute.

We appreciate Reviewer 1's engagement with the manuscript and calling our attention to places where our analysis procedures can be clarified.

R1.2 The national database constructed between 1999 and 2020 is a novel and useful resource. I wonder about the start date of 1999 and why this date was chosen. A justification for both the start and end date would be helpful.

We clarify on lines 517-521 why we select these years, writing: “We utilize data from the full temporal period available in the most recent publication of the ICS, 1999 through 2020. 1999 is the first year for which the National Wildfire Coordinating Group provides raw data from which the ICS is produced. 2020 represents the most recent year through which the ICS have been cleaned.”

R1.3 The findings are original and constructively inform and drive forward the conversation needed for how we respond to and recover from wildland fires. The dominant finding, and one that is a bit buried, is that immobility is the most common response to wildfire (which is similar to other hazards). I might encourage them to highlight this a bit more as a key takeaway. At the granular level, their data suggest that extreme, outlying wildfire events drive the majority of building destruction—the Camp Fire serving as the most destructive—and that wildfire building destruction drives increased out-migration only at the highest severity levels—again with the Camp Fire serving as the example. In short, they did not find population level migration changes.

We agree with this assessment and have added text to highlight immobility in several places throughout the manuscript: (1) In the Introduction, we have expanded our discussion of immobility in paragraphs starting on lines 41 and 54. There are many reasons to expect that

immobility may be observed in response to wildfire destruction, and we now lay the groundwork for this finding more clearly. (2) We pick this immobility framing back up in a new paragraph introducing the Discussion (starting on line 302) and affirm from the outset that immobility is the most commonly observed response to destructive wildfires (lines 310-312). (3) Further in the Discussion when describing our immobility finding (paragraph starting on line 375), we have added additional context for this finding, emphasizing the importance of identifying climate-related immobility.

R1.4 Finally, they provided support for hypothesis 2 with evidence that indirect wildfire effects had an impact on out migration two years after the event. I would caution the interpretation of results here. While the authors document the trend, they have no support about the motivations behind the migration (lines 212-213).

We have adjusted this language to distinguish between observed results and how we use our theoretical framing of direct/indirect effects to interpret these results.

R1.5 The authors analyze four subsets of the decile, but do not describe this approach at the outset. The description comes under the methods section (458-462). It would make it easier on the reader to provide some guidance about this approach on lines 158-159 before moving into the results description. In other words, a description like “our analysis proceeds in four stages and explores the deciles in the following manner....” Would make it easier for the reader to track.

We have added in a short description of the wildfire severity-based subsets on lines 194-197.

R1.6 The authors also count “structures” in the ICS dataset. Structures include both houses and outbuildings. While the ICS counts provide a direct measure of hazard impact, they do not separate houses from outbuildings. This may contribute to overstating impacts on houses if the figures are conflated. Likewise, a bit more careful attention to the language throughout and definitions up front would be useful. The terms buildings, houses and structures are used interchangeably, and greater precision focused on “structures” and what they are exactly is likely warranted.

Throughout the text, we have edited our language so that we only use the term “structures” when referring to estimates from the ICS data. Additionally, on lines 515-517 we have added a description of all building types that are considered “structures” in the ICS structure loss counts.

R1.7 Finally, the authors may have seen the recent publication by Radeloff et al. 2023 in Science on rising wildfire risk to houses in the US, especially in grasslands and shrublands. This is an informative and useful publication that also speaks to the trends in this article.

Thank you for calling our attention to this article - we incorporate it throughout the manuscript (citation number 2).

R1.8 I really enjoyed reading this article and learned a lot from it. It was well written, well structured and I very much look forward to seeing it in print. Toddi Steelman

We appreciate Reviewer 1’s interest in our work, thank you.

Reviewer 2

R2.1 Thank you for the opportunity to review this paper, which is well-designed and well-written. The authors examine how wildfires have impacted human migration in the continental US between 1999 and 2020, focusing on the 10% most destructive wildfires in that period. They highlight two pathways for migration differences: direct (damage-driven) and indirect (changes in residential preferences and/or capabilities). I believe this to be a strong paper and recommend that the authors be given the chance to submit revisions toward future publication.

We appreciate Reviewer 2's emphasis on situating our study more deeply within existing scholarship and believe that our subsequent revisions have improved the manuscript's contribution to the literature.

R2.2 My primary comment is that I wish the article were more clearly rooted in the literature, which could help to contextualize and make sense of some of the current findings. It may be true that wildfires differ from other hazards in significant ways, in which case, the authors should explain why. Further, some discussion of broader migration trends for other reasons—such as people leaving fire-prone California because of cost-of-living—seems like important context here.

To address the first component of this comment, we've made the following changes: (1) In the Introduction, we have added new text between lines 41 and 70. The literature reviewed here speaks more specifically to migration observed following sudden-onset environmental hazards, which are those most comparable to wildfires. It further develops the concept of immobility and contextualizes population dynamics in the context of wildfire-prone places in the U.S. (2) In the Introduction, we reference our framing approach of "direct" and "indirect" hazard impacts on migration to Hoffman et al. 2020 (line 83). (3) We added a new paragraph at the outset of the Discussion (starting on line 302) to more clearly situate our work within existing literature. (4) In the Discussion, we have added additional text on immobility (lines 384-388). (5) Finally, we have added additional text in the Discussion (paragraphs starting on lines 439, 450, 473, and 484) that discusses future research needs within the broader context of existing environmental migration and wildfire literatures.

To address the second component of this comment, we include additional text that speaks to broader wildfire-population dynamics (starting on line 54). Specifically, we address the environmental amenity pull of migration into fire-prone places, citing Winkler and Rouleau 2020 as well as Hammer et al. 2009. We also now reference new research by Greenberg et al. (forthcoming in *PNAS*) which describes how rising housing costs in California's urban cores act as a contributing push factor on migration into fire-prone places, as residents seek more affordable housing.

R2.3 That said, the main finding of the paper—that people move away when their area experiences significant damage from wildfire—may be important, but it's not particularly surprising. It seems to me that others have addressed the issue of hazard (including wildfire) migration with greater attention to some explanatory factors, such as Winkler and Rouleau 2020 (<https://doi.org/10.1007/s11111-020-00364-4>). I think the authors need to make a better case for why this pretty intuitive finding is novel and/or important.

We have made a number of changes to clarify our study's contribution to existing research on wildfire-related migration, with special attention to Winkler and Rouleau 2020. (1) We have

moved up and expanded a paragraph that was previously located in our Methods section, which describes our study's methodological advances relative to most comparable existing studies in this area (paragraph starting on line 400). To briefly summarize, our study offers event severity stratification, a more granular spatial scale, and improved causal identification through our quasi-experimental research design. (2) We include a new supplementary analysis (Appendix 2) examining the spatial scale of destructive wildfires relative to spatial units in which wildfires occur. We show that the county scale (which is used by both Winkler and Rouleau 2020 and DeWaard et al. 2023, the most comparable studies to our manuscript) is very coarse relative to the burn area of destructive wildfires. Our use of census tracts is therefore an important methodological improvement to the study of wildfire-related migration. (3) Winkler and Rouleau 2020 rely on FEMA disaster declarations to identify fire-affected counties. As a result, this study does not differentiate the level of impact severity across wildfire events. Our use of wildfire structure loss data allows us to parse different types of wildfires (more and less destructive events), thereby contributing to environmental migration literature focused on non-linear effects and the importance of severity thresholds at which effects are observed. This area of research was identified as particularly important by two recent systematic reviews, Hoffman et al. 2020 and Kaczan and Orgill-Meyer 2020.

To emphasize the novelty and importance of this work, we have made the following changes: (1) We have added new text focused on immobility in the context of environmental hazards to both the Introduction and Conclusion (paragraphs starting on lines 41 and 375). Identifying immobility in many cases will mean identifying null effects, which traditionally are less likely to be published. Yet, the focus on immobility is an emerging and important direction for environmental migration research, as identified by Schewel 2020, Cundill et al. 2021, and Zickgraf 2021. (2) In our deeper engagement with literature, we highlight that existing studies of sudden-onset hazards (those most similar to wildfires) find both immobility and heightened out-migration outcomes. Our empirical strategy of stratifying wildfires by level of destruction allows us to clarify that severity is an important factor in determining why some sudden-onset events result in migration changes while others do not.

With regard to the point that our results are not especially surprising, we would argue that, given the heterogeneity of migratory responses to comparable sudden-onset hazards (hurricanes, tsunamis, floods), it is not obvious that we would have observed the results we did in the context of wildfires. Further, while our results may seem post facto intuitive to academic researchers working in this space, in public discourse, we continue to observe what Hunter et al. 2015 has referred to as the "maximalist position" on environmental migration, which posits that residents will simply move away from hazardous places as they become more hazardous. Our results demonstrating that in fact *immobility* is a more common response to wildfire paint a very different picture, contrasting to the more maximalist climate migration narratives that have become common in media. To this end, we believe that rigorous empirical evaluation of wildfire-immobility dynamics is especially important.

R2.4 Is the implication that migration increases in highly impacted areas because more people lost their homes and left, or because there is a threshold of surrounding damage that people are less likely to tolerate, regardless of the survival of their home? The authors may

be unable to answer this with their data, but the two possibilities speak to two very different processes.

We are unable to distinguish between these two possibilities with our data. We write starting on line 348, “Our research design ultimately does not allow us to distinguish between residents moving away because their own dwellings were destroyed, because their local environment experienced high levels of destruction, or a combination of both. These possible pathways should be investigated in future research with qualitative methods focused on migration decision-making.”

R2.5 Finally, there are some real equity concerns around this topic that are mostly ignored by the authors. I’d like to see this addressed more explicitly. The authors do briefly address it in the limitations, regarding problems with the data itself, but it is also relevant to the findings. For example, all in-migration is not the same; there is an important difference between people returning home after disaster (what the authors attribute to “recovery migration” after the Camp Fire), and others taking advantage of low property prices after a fire.

We especially appreciate this recommendation and have made the following revisions in the Discussion to more clearly identify equity considerations for this research: (1) The paragraph starting on line 439 describes a limit to the CCP migration data being its lack of demographic decomposability, meaning that we are not able to examine demographic subgroup-specific migration trends (e.g. across differences in race, ethnicity, or nativity) related to questions of disproportionate impact. (2) The paragraph starting on line 450 considers the fact that the CCP migration data do not include residents without a Social Security Number, and therefore underrepresent financially vulnerable residents. We reference two articles which suggest that specific groups may be especially susceptible to wildfire impacts (McConnell and Braneon 2024 and Méndez et al. 2020) and make the point that new data are needed to investigate how wildfire impacts affect these residents. (3) The paragraph starting on line 473 speaks to the second point in this comment regarding post-wildfire in-migration. We highlight the importance of further research focused on in-migration dynamics that can distinguish between returning residents and new arrivals (which we cannot do with our study design). We also reference housing dynamics that may be relevant to this line of inquiry, including housing affordability and post-wildfire gentrification. (4) Finally, all existing literature on wildfire and migration that we were able to find and build our work on is geographically based in the U.S. Given that wildfires are a global phenomenon, we see this as a serious limitation of existing scholarship and conclude the paper with a call to study wildfire-mobility relationships in different countries, especially those that are agriculturally dependent (paragraph starting on line 484). If there are additional equity considerations that we have overlooked and can be incorporated, we would be glad to make additional revisions.

R2.6 Again, the authors should be commended for a thoughtfully designed study and well-written paper. I believe with a little more work it can provide a valuable contribution to the existing literature on wildfire impacts and climate migration.

Thank you again for your time reviewing our work.

REVIEWERS' COMMENTS

Reviewer #1 (Remarks to the Author):

I have reviewed the revised manuscript and am generally satisfied with the revisions. The authors have been attentive to the reviewer critiques, including my own. The article is better for these corrections, especially the stronger contextualization within the environmental migration and hazards literature and emphasis on the dominant finding of immobility. I make three final, minor recommendations below:

Line 46: modify liquidity with "financial" to read "from financial liquidity constraints..."

Line 90: first mention of structures (I believe). While definition of structures occurs in methods on line 517, the definition should come earlier to help the reader understand what they mean by structures throughout the article. The definition at the end is too late by my read. I recommend you define structures at the first mention of them.

Finally, the destructive wildfires in Lahaina, HI may be a further opportunity to test the same effect observed in the Camp Fire. It might be useful to signal this as an area of further research for two reasons-- 1) the opportunity to look outside the contiguous US; 2) further verify or not the same migratory effects observed in the larger data set/Camp Fire.

Response to Reviewers:

Reviewer 1 Comment 1: Line 46: modify liquidity with "financial" to read "from financial liquidity constraints..." We have made this revision.

Reviewer 1 Comment 2: Line 90: first mention of structures (I believe). While definition of structures occurs in methods on line 517, the definition should come earlier to help the reader understand what they mean by structures throughout the article. The definition at the end is too late by my read. I recommend you define structures at the first mention of them. We now define "structures" in the Introduction on lines 105-106.

Reviewer 1 Comment 3: Finally, the destructive wildfires in Lahaina, HI may be a further opportunity to test the same effect observed in the Camp Fire. It might be useful to signal this as an area of further research for two reasons-- 1) the opportunity to look outside the contiguous US; 2) further verify or not the same migratory effects observed in the larger data set/Camp Fire. We have added the following sentence to the Discussion: "Additionally, research in this area could productively examine whether more recent extreme events and events outside of the contiguous U.S., such as the 2023 Maui Fire in Hawaii, have similar migration effects as those found in this analysis" (lines 385-387).